# Investigation of Tannic Acid Crosslinked PVA/PEI-Based Hydrogels as Potential Wound Dressings with Self-Healing and High Antibacterial Properties

**DOI:** 10.3390/gels10110682

**Published:** 2024-10-23

**Authors:** Nimet Rumeysa Karakuş, Serbülent Türk, Gamze Guney Eskiler, Marat Syzdykbayev, Nurbol O. Appazov, Mahmut Özacar

**Affiliations:** 1Department of Biomedical Engineering, Institute of Natural Sciences, Sakarya University, 54187 Sakarya, Türkiye; nmtrmys.06@gmail.com; 2Biomaterials, Energy, Photocatalysis, Enzyme Technology, Nano & Advanced Materials, Additive Manufacturing, Environmental Applications and Sustainability Research & Development Group (BIOENAMS R & D Group), Sakarya University, 54050 Sakarya, Türkiye; serbulentturk@sakarya.edu.tr; 3Biomedical, Magnetic and Semiconductor Materials Application and Research Center (BIMAS-RC), Sakarya University, 54187 Sakarya, Türkiye; 4Department of Medical Biology, Faculty of Medicine, Sakarya University, 54100 Sakarya, Türkiye; gamzeguney@sakarya.edu.tr; 5Laboratory of Engineering Profile “Physical and Chemical Methods of Analysis”, Korkyt Ata Kyzylorda University, Aiteke bi Str., 29A, Kyzylorda 120014, Kazakhstan; marat.1980@mail.ru; 6KazEcoChem LLP, D.Konaev Str. 12, Astana 010010, Kazakhstan; 7Department of Chemistry, Faculty of Science, Sakarya University, 54050 Sakarya, Türkiye

**Keywords:** hydrogel, tannic acid, anti-freezing, drug release, self-healing, antibacterial properties

## Abstract

This study developed hydrogels containing different ratios of TA using polyvinyl alcohol (PVA) and polyethyleneimine (PEI) polymers crosslinked with tannic acid (TA) for the treatment of burn wounds. Various tests, such as scanning electron microscopy (SEM), Fourier transform infrared spectroscopy (FTIR), swelling, moisture retention, contact angle, tensile strength, the scratch test, antibacterial activity and the in vitro drug-release test, were applied to characterize the developed hydrogels. Additionally, the hydrogels were examined for cytotoxic properties and cell viability with the WST-1 test. TA improved both the self-healing properties of the hydrogels and showed antibacterial activity, while the added gentamicin (GEN) further increased the antibacterial activities of the hydrogels. The hydrogels exhibited good hydrophilic properties and high swelling capacity, moisture retention, and excellent antibacterial activity, especially to *S. aureus*. In addition, the swelling and drug-release kinetics of hydrogels were investigated, and while swelling of hydrogels obeyed the pseudo-second-order modeling, the drug release occurred in a diffusion-controlled manner according to the Higuchi and Korsmeyer–Peppas models. These results show that PVA/PEI-based hydrogels have promising potential for wound dressings with increased mechanical strength, swelling, moisture retention, self-healing, and antibacterial properties.

## 1. Introduction

The skin, the body’s greatest organ, behaves as a protective barrier against pathogens and provides defense. Open wound surfaces are highly susceptible to infection by bacteria [1,2]. In biomedical applications, reducing infection risks, promoting wound healing, and supporting tissue repair are crucial challenges. Hydrogel wound dressings are considered to be an easier and more effective method for wound treatments [3]. Consequently, hydrogel wound dressings with crosslinked polymeric chain networks and hydrophilic functional groups are being developed [4]. These three-dimensional polymeric chain networks allow the hydrogel to sorb and hold high quantities of water without dissolving [5].

Desired properties include gas permeability, maintaining a moist medium in the wound region, flexibility, capability to absorb excess exudates, protection against bacteria by reducing the possibility of wound infection due to its antibacterial properties, ease of removal and replacement without damaging the wound area, good mechanical strength, and biocompatibility [6,7].

The process of wound healing is a complex and coordinated sequence of events that initiates as soon as a tissue is damaged by various causes [7]. It encompasses four interrelated phases, namely hemostasis, inflammation, proliferation, and remodeling (maturation) [8,9]. The inceptive phase, known as hemostasis, involves the formation of a platelet plug by platelets and fibrins to achieve coagulation and stop bleeding if present in the wound area [10,11]. This phase is followed immediately by the inflammatory stage, which is influenced by numerous internal and external factors [12]. Concurrent with hemostasis, the inflammatory phase involves the participation of situated and newly formed cells of the inborn and acquired immune system. Its primary role is to mitigate the risk of infection by eliminating residual and pathogenic substances from the wound area, with chemokines playing a central role in this process [13,14]. Additionally, inflammatory cells serve as a crucial source of growth factors and cytokines that initiate the proliferative phase [15].

The subsequent phase is proliferation, during which secretory factors increase vascular permeability, and cytokines and growth factors contribute to the recruiting of endothelial, epidermal, and dermal cells to the wound site [16]. This phase also involves fibroblasts and collagen synthesis, initiating angiogenesis by promoting the generation of granulation tissue and facilitating the progression to the final phase, namely tissue remodeling [17,18]. Approximately 1–3 weeks after the injury, fibroblasts differentiate into myoblasts, and the synthesis of type I collagen increases, providing the healing of the extracellular matrix (ECM) and the initiation of tissue remodeling [12,19]. The newly formed network of blood vessels evolves into a ripe tissue structure with restricted structural strength, ultimately forming scar tissue [20]. This intricate process relies on the body’s utilization of complex biochemical and cellular mechanisms to orchestrate wound healing [21], culminating in the completion of the wound-healing process.

Hydrogel dressings designed to promote the wound-healing process can be prepared using various methodologies. These approaches fall within three categories, namely physical, chemical, and UV irradiation [22]. A careful selection of polymers used for hydrogels that can be synthesized via physical and chemical crosslinking is essential. These polymers, categorized as natural and synthetic, serve as the primary constituents in hydrogel production [23].

Polyvinyl alcohol (PVA), developed by Hermann and Haehnel in 1924, stands as one of the earliest synthetic polymers [24]. Its water-soluble nature has garnered significant attention in the chemical and medical sectors owing to its distinctive attributes, such as biocompatibility, adhesion to diverse surfaces, non-toxic properties, and biodegradability [25]. Meanwhile, polyethylenediamine (PEI), a synthetic polymer categorized under the polycation class, features flexible polymeric chains and exhibits considerable promise for biomedical applications due to its high sensitivity, biocompatibility, and durability [26]. Notably, its crosslinking property has made it a preferred choice in the formulation of hydrogel solutions [27]. The recent focus in bioengineering has been on the advancement of self-healing hydrogels, which possess unique dynamic responsiveness [7]. This development aims to enhance the structural integrity of hydrogels, making them more resistant to external impacts.

Tannic acid (TA) is a native polyphenol present in various plants, known for its biocompatible properties and wide prevalence in nature [28]. It serves as an effective crosslinker in polymer bonding, owing to its ability to induce physical or chemical binding. Additionally, TA exhibits notable antibacterial properties, making it a valuable component in the treatment of burn wounds [29].

Due to its non-toxicity, environmental friendliness, and good biocompatibility, PVA has been widely used in various applications, such as drug delivery systems, tissue engineering, and hydrogel wound dressings [30,31]. Despite the advantages mentioned, the poor mechanical properties and lack of antibacterial properties of PVA hydrogels limit their further application as wound dressings. To overcome these limitations, researchers have often prepared PVA-based antibacterial hydrogels using different crosslinkers, as well as metal nanoparticles or other agents [31]. Many methods, such as formic acid, boric acid, various crosslinking agents, the wet spinning technique, and freeze–thaw cycles have been used to crosslink PVA chains while preparing PVA hydrogels [32]. Cells, drugs, Ag nanoparticles, or plant-based natural components have been used to prepare wound dressings with effective antibacterial properties [33]. While preparing hydrogels with PVA and various polymers, TA has been investigated in different studies for the green synthesis of Ag and other metal particles used to impart antibacterial properties, crosslinking of hydrogels, and contribute to antibacterial properties [32,34,35]. In this study, TA acid was used to crosslink polymer chains in hydrogels prepared with PVA and PEI and to exhibit the antibacterial properties of hydrogels. Unlike similar studies in the literature, gentamicin (GEN) was used together with TA to increase the antibacterial properties of hydrogels by showing a synergistic effect in this study.

The advancement of antibacterial hydrogel wound dressings was the focus of this study, achieved through the crosslinking of PVA and PEI polymers in the presence of TA. Samples with self-healing properties for controlled drug release were obtained by incorporating GEN. Various characterization tests were conducted on the samples, including Fourier transform infrared (FTIR) analysis, to examine the desired functional groups of PVA/PEI hydrogels, tests for swelling abilities and moisture retention capacities, and a contact angle test to determine hydrophilic properties. Tensile tests were performed to assess mechanical strength, while SEM and FESEM-EDS analyses were conducted to understand surface morphologies and elemental composition. Antibacterial tests against *E. coli* and *S. aureus* bacteria were executed to determine the materials’ antimicrobial capabilities, with an additional cell viability analysis performed through WST-1 tests. Scratch tests were performed to evaluate the self-healing abilities, and the swelling and drug-release kinetics of the hydrogels were studied.

## 2. Results and Discussion

### 2.1. FTIR Spectroscopy Analysis

The FTIR analysis was conducted to identify the characteristic absorption bands belonging to the functional groups of PVA/PEI-based hydrogel samples with and without drug addition along with TA. This analysis was essential in the elucidation of the bonds and interactions between the polymers and TA. The large bands observed between 3550 and 3200 cm^−1^ are related to the stretching O–H from the intermolecular and intramolecular hydrogen bonds’ PVA forms [36]. While the broad peak in the region of 3550–3100 cm^−1^ is characteristic of the –OH stretchings of the phenolic and methylol groups of TA, the small peaks near 2900 cm^−1^ are due to aromatic C–H stretching vibrations [37,38]. The broad peaks, observed between 3200–3600 cm^−1^ in all hydrogel samples, delineate the absorption band of the –OH groups of PVA polymer and the phenolic –OH groups of TA, confirming the presence of TA bonds in PVA/PEI-based hydrogel samples [39]. Additionally, the absorption bands around 3300 cm^−1^, ascribed to N-H stretching vibrations, confirm the existence of the amine groups of the PEI polymer [40]. In addition, the broad bands in the 3600–3200 cm^−1^ region of the FTIR spectra are also associated with the water physically adsorbed by the samples. Furthermore, the peaks around 2960 cm^−1^ in the sample analysis were associated with C-H stretching in the CH_2_ and CH_3_ groups of PVA and PEI polymers [41]. The peaks originating from C=O groups and C=C stretching found in the structure of PVA and PEI polymers and TA molecules are located around 1700–1750 cm^−1^. The peaks originating from these groups are shifted slightly and appeared in the 1644–1659 cm^−1^ regions due to the interactions of the polymer chains and the crosslinking between the polymer chains’ TA molecule forms. This change is most likely due to the environmental change of the C=O groups in the hydrogel structures [36,37,42]. The observed bands around 1650 cm^−1^ were determined to be related to C=O stretching vibrations [43], indicating the presence of GEN and C=O stretching. Analysis of the peaks between 1500 and 1000 cm^−1^ revealed the presence of C-H stretching at 1390 cm^−1^ and a C-O-C bond at 1135 cm^−1^ [44]. Additionally, the stretching vibrations of the C-O bond were characterized by peaks around 1070 cm^−1^ [45]. These results (Figure 1A,B) confirm that PEI successfully formed a polymeric chain with PVA due to the chemical crosslinking reaction provided by TA. To assess the influence of the GEN drug in the FTIR analysis, the peak values from the literature were examined, ultimately indicating that the band shifts in the peaks signified the presence of GEN [46].

### 2.2. Swelling Ratio Test

For effective wound healing, a hydrogel dressing with a high absorption capacity is needed [47]. A swelling test was conducted on the samples to assess this. The swelling capacities of hydrogels are contingent on the quantity of TA utilized as a crosslinker. The crosslinkers bind the polymers, reducing the hydrogels’ pore size. Therefore, it is generally accepted that the swelling capacity decreases as the crosslinker ratio in the hydrogel increases [48]. Figure 2 illustrates the results of the swelling test performed on PVA/PEI-based hydrogel samples. The hydrogel samples reached equilibrium after 24 h. Upon examination of the graphs, it is evident that the sample demonstrating the best swelling capacity is drug-added PVA/PEI/GEN/TA1. Surprisingly, the ionization of TA in the solution seems to have mitigated the anticipated impact of increased crosslinker density on the swelling capacity. Notably, the crosslinked sample exhibited a significantly improved swelling capacity compared to the sample without it. It was observed that the addition of TA increased the swelling rate compared to the hydrogel without TA. It was evaluated that further increasing the TA content in the hydrogel provided non-covalent interactions for the hydrogels by forming more hydrogen bonds between the polymer chains, thus preventing further swelling of the hydrogels. Since the prepared hydrogels have sufficient swelling rates to keep the wound surface moist and facilitate exudate absorption, they may be potential materials for wound dressings. The results found are quite consistent with previous studies showing that TA provides a certain swelling rate to the hydrogels, that the swelling rates generally tend to decrease non-linearly with the increase in the amount of TA, and that the decrease in the swelling rates is not regular with the amount of TA [49,50].

To understand the swelling mechanism of the prepared hydrogels, the Peppas, pseudo-first-order, and pseudo-second-order kinetic models were used to evaluate the experimental swelling data. It is important to understand the diffusion mechanisms of the hydrogels to be used as drug carriers. Diffusion involves the movement of a water/drug solvent into the spaces between the hydrogel networks, which leads to expansion between the crosslinked chains. One of the models used to determine the water absorption of hydrogels and their swelling ratio at certain time intervals is the Peppas equation, and its mathematical expression is as follows [51,52,53]:(1)$$\frac{S_{t}}{S_{\mathrm{eq}}}={\mathrm{kt}}^{n}$$

The linear form of the Peppas equation is
(2)$$ln\frac{S_{t}}{S_{eq}}=\ln k+nlnt$$
where S_t_ and S_eq_ are the amounts of water taken up by the hydrogel at the swelling time and at equilibrium, k is the swelling rate constant, and n is the swelling exponent that is an indicator of the water penetration movement. The values of n and k are obtained from the slope and intercept of ln S_t_/S_eq_ vs. lnt plot, respectively. The value of n gives an insight into the water absorption and swelling mechanism of the hydrogel. If the value of n is between 0.45 and 0.5, it is evaluated that the water absorption follows a diffusion-controlled Fickian kinetics, while the values of n between 0.5 and 1.0 indicate that the hydrogel shows a non-Fickian diffusion mechanism in which chain relaxation also contributes to water absorption [52,53].

To investigate the swelling kinetics of hydrogels, pseudo-first-order (Fick’s model), pseudo-second-order (Schott model), and the Peleg equations were used. The pseudo-first-order, pseudo-second-order, and Peleg equations are given in Equations (3)–(5), respectively [53,54,55,56].
(3)$$\frac{{dS}_{t}}{dt}=k_{1}\left(S_{eq}-S_{t}\right)$$
(4)$$\frac{{dS}_{t}}{dt}={k_{2}\left(S_{eq}-S_{t}\right)}^{2}$$
(5)$$\frac{{dS}_{t}}{dt}=\frac{{k’}_{1}}{{\left({k’}_{1}+{k’}_{2}t\right)}^{2}}$$

When the pseudo-first-order, pseudo-second-order, and Peleg equations are integrated at the boundary conditions (S = S_0_, t = t_0,_ and S = S_t_ at t = t), the linear pseudo-first-order, pseudo-second-order, and Peleg equations obtained are Equations (6)–(8), respectively.
(6)$$ln\left(S_{eq}-S_{t}\right)={lnS}_{eq}-k_{1}t$$
(7)$$\frac{t}{S_{t}}=\frac{1}{k_{2}S_{eq}^{2}}+\frac{1}{S_{eq}}t$$
(8)$$\frac{t}{S_{t}-S_{0}}={k’}_{1}+{k’}_{2}t$$
where S_eq_, S_t_, and S_0_ denote the swelling amount at equilibrium, at time t, and t = 0, respectively. k_1_ and k_2_ are the pseudo-first-order and pseudo-second-order kinetic rate constants, respectively. k’_1_ and k’_2_ are the kinetic constant and a characteristic constant of the Peleg model, respectively.

To better understand the swelling kinetics, the results obtained by applying different kinetic models to the experimental swelling data are shown in Appendix A. The kinetic parameters and correlation coefficients (r^2^) found for different kinetic equations are given in Table 1. The best-fitting kinetic equation for the experimental swelling kinetic data was evaluated with the correlation coefficient values. When the r^2^ values in Table 1 are examined, the highest r^2^ values were obtained for the pseudo-second-order equation and followed the r^2^ values of the Peleg equation. It is seen that the pseudo-second-order kinetic model provides the best correlation for the swelling process, whereas the Peleg model also fits the experimental data well. Therefore, the model that best obeys the experimental swelling data is the pseudo-second-order equation.

### 2.3. Water Retention Rate Test

The capacity of hydrogels to preserve a dewy medium and absorb wound exudates, known as water retention, is a crucial property [57]. A water retention test was conducted on the samples by weighing them at predetermined time intervals after drying at room temperature using a precision scale. The results, as depicted in Figure 3, indicated that the water retention capacities of the samples decreased over time until reaching an equilibrium profile. Comparing the water retention capacities of samples with and without the addition of GEN revealed that the PVA/PEI/GEN/TA1 and PVA/PEI/GEN/TA2 samples exhibited the highest increase in water retention capacity. Although PVA/PEI/GEN/TA4 appeared to have the highest water retention capacity based on the graph, the notable improvement in the moisture retention capacity of the PVA/PEI/GEN/TA2 sample after drug addition is noteworthy.

The PVA/PEI/GEN/TA2 sample is promising for clinical studies in wound-dressing applications due to its increased water retention capacity after drug loading. Compared to the hydrogel samples in the existing literature, it is believed that hydrogel samples with an equivalent moisture retention capacity have been developed [58]. The ability to retain moisture provides a significant advantage for hydrogels, allowing them to dry more slowly and be used for extended periods. To enhance the moisture retention capacity of hydrogels, materials with superior moisture retention properties, such as hyaluronic acid (HA), can be incorporated into the solution [59].

It has been reported in the literature that increasing the crosslinking density increases the water retention capacity of hydrogels [60]. It has also been noted that the water retention of hydrogels can be also affected by the existence of van der Waals forces and H-bonds between the hydrogel and water molecules [61,62]. The presence of the –COO– group in the polymer chains of hydrogels, which provides low water loss, contributes to an exceptional water retention capacity by enhancing the polymer network’s affinity for water molecules. Therefore, incorporating TA molecules containing the –COO– group into the solution can enhance its moisture retention capability.

### 2.4. Contact Angle Results

In contact angle measurement, the reference degree accepted for the angle is 90° [63]. The angle between DW and the hydrogel samples allows for determining the hydrogel’s water absorption ability. A contact angle smaller than 90° (23–33°) indicates hydrophilic properties (Figure 4). Analysis of GEN drug-added PVA/PEI samples revealed excellent hydrophilic properties. Similarly, drug-free hydrogel samples exhibited good hydrophilic properties, with contact angle values of 17–28°. These findings suggest that both the drug-added and drug-free samples possess favorable hydrophilic properties.

### 2.5. Mechanical Test

The mechanical features of hydrogels were analyzed via a tensile test [64], and the results, calculated in MPa, are presented in Table 2. Appendix A illustrates that the PVA/PEI-based hydrogel samples with a low concentration of TA exhibited good mechanical strength under the applied tensile force. However, an increase in TA resulted in decreased mechanical strength. Samples PVA/PEI/TA4 and PVA/PEI/GEN/TA4 were not tested, as they fractured during manual handling before the test. The PVA/PEI/GEN/TA3 sample ruptured at the beginning of the tensile process. Notably, samples without TA addition demonstrated the highest mechanical strength.

It is believed that the ionization of the TA crosslinker in the solution adversely affected the mechanical strength, making the samples more brittle than anticipated. Effective hydrogel wound dressings require not only increased mechanical strength but also flexibility. Considering the results, the PVA/PEI/TA0 and PVA/PEI/TA1 samples, both with and without the addition of drugs, are deemed suitable for hydrogel wound dressings. Modification is necessary to enhance the mechanical strength of the remaining samples for potential use in wound dressings.

### 2.6. SEM Results

The study examined the morphological structures of freeze-dried hydrogel samples [65]. The porous nature of hydrogel dressings offers advantages in terms of absorbing wound exudate, retaining water, preventing wound infection, and promoting wound healing by maintaining a moist environment [66]. The concentration of crosslinking agents, such as PEI and TA, influences the porous structure, with higher concentrations leading to fewer pores due to increased intermolecular bonding [27]. Notably, SEM micrographs of hydrogel samples in Figure 5 revealed that the porous structure did not consistently follow an expected trend [67]. Specifically, the PVA/PEI/TA1, PVA/PEI/TA3, and PVA/PEI/GEN/TA2 samples exhibited a porous structure, while the others displayed nonporous and irregular morphologies. The variations in the surface properties of TA-added PVA/PEI-based hydrogels may be attributed to the strong crosslinking capabilities of PEI [68].

### 2.7. Antibacterial Activity Test

*E. coli* and *S. aureus* bacteria near a wound may lead to inflammation and hinder wound healing [69]. Hence, developing antibacterial wound dressings is crucial to facilitate the healing process. Upon examination of the samples incubated at 37 °C for 24 and 48 h in cultured media containing *E. coli* and *S. aureus* bacteria, it was observed that all exhibited significant antibacterial properties, as illustrated in Figure 6. The notable zone diameters of both the drug-free and drug-added hydrogel samples indicated the antibacterial properties of PEI, a synthetic polymer commonly used in hydrogel synthesis.

### 2.8. Self-Healing Properties

Enhancing the self-healing properties of hydrogel wound dressings presents a viable alternative for mitigating material aging and wear caused by impacts. This attribute is first ascribed to the existence of electrolytes and covalent, ionic, and polar functional groups, which facilitate the formation of hydrogen bonds [70]. Evaluation of the self-healing capability involved creating scratches on the samples using a disinfected razor blade, as illustrated in Figure 7. The images captured after 5 and 15 min and after self-healing and storage at −20 °C revealed that PVA/PEI/TA0 and PVA/PEI/TA1 exhibited completely self-healing properties without external stimuli. This observation signifies a promising advancement in the development of self-healing wound dressings.

### 2.9. Drug-Release Profiles

The amount of drug released from gentamicin-added PVA/PEI-based hydrogel samples was determined by creating a standard graph (Appendix A) using taken measurements for different concentrations. The drug-release ratios of TA-containing wound dressings are with a faster release within the first 6 h between 9.5–11.5%, then reached a slow release equilibrium, and after 48 h, the cumulative drug-release rates were approximately 14.30–17%. In contrast, the drug-release rate of the dressing without TA was 13% in the first 6 h. Then, the cumulative drug-release ratio in the slow-release equilibrium reached approximately 18.40%. The initial rapid drug release followed by a slow drug release will provide effective inhibition and clearance for bacteria in treatment by wound dressing. The results, depicted in Figure 8, revealed that TA addition in PVA/PEI/GEN/TA1, PVA/PEI/GEN/TA2, PVA/PEI/GEN/TA3, and PVA/PEI/GEN/TA4 hydrogels facilitated a more controlled release of GEN compared to the TA-free PVA/PEI/GEN/TA0 hydrogel. This observed behavior can be ascribed to the existence of the crosslinker, as TA enhances the polymer bond, leading to a controlled drug release by reducing the pore size [71]. Furthermore, it was determined that the strong hydrogen bonds generated via the numerous -OH groups of the TA molecule with the -NH_2_ groups in the GEN structure significantly contribute to the controlled drug release. In conclusion, incorporating TA in PVA/PEI-based hydrogel dressings offers a promising wound healing and repair approach, providing longer and controlled drug release.

In order to better evaluate the drug-release mechanisms of hydrogels, experimental drug-release data can be correlated with drug-release kinetic models. Zero- and first-order, Higuchi, Korsmeyer–Peppas, and Hixson–Crowell equations have been widely used to examine the drug-release kinetics. The mathematical expressions of these equations are given below, respectively [51,72,73].
Zero-order equation C_t_ = C_0_ + k_0_t(9)
First-order equation log C_t_ =log C_0_ − (k_1_/2.303) t(10)
Higuchi equation C_t_ = k_H_t^1/2^(11)
Korsmeyer–Peppas equation log (C_t_/C_∞_) = log k_KP_ + nlog t(12)
(13)$$\mathrm{Hixson}–\mathrm{Crowell}\;\mathrm{equation}\;W_{0}^{1/3}-W_{t}^{1/3}=k_{HC}t$$
where C_t_ is the concentration of the drug released at time t, C_0_ is the initial concentration of the drug at time t = 0, C_∞_ is the concentration of the drug released after time ∞, W_0_ is the initial amount of drug in the pharmaceutical dosage form (amount of drug remaining at time 0), W*_t_* is the remaining amount of drug in the pharmaceutical dosage form at time t, and k_0_ is the zero-order rate constant. k_1_ is the first-order rate constant, k_H_ is the Higuchi dissolution constant, n is the drug-release exponent, k_KP_ is the Korsmeyer release rate constant, k*_HC_* is the Hixson–Crowell constant describing the surface volume relation, and t is the time.

For a better understanding of the drug-release kinetics of hydrogels, the results obtained by applying various kinetic models to the experimental GEN release kinetic data are shown in Appendix A, and the kinetic parameters and r^2^ found for each kinetic equation are given in Table 3. When the r^2^ values in Table 3 are compared, it is seen that the r^2^ values of the Korsmeyer–Peppas and the Higuchi model for GEN release from hydrogels are quite close to each other and have higher values than that of the other models. It can be interpreted that the prime mechanism of GEN release from hydrogels is a diffusion-controlled release mechanism. Once it has been determined that the prime GEN release mechanism is diffusion controlled from the Higuchi model, then it gains importance that the drug release obeys which type of diffusion. The n value in the Korsmeyer–Peppas model can be used to characterize different diffusion mechanisms in drug release. It is seen that the values of the drug-release exponent or the diffusion exponent (n) in Table 3 are generally in the range of 0.45–0.5, which means that the GEN release from hydrogels follows diffusion-controlled Fickian kinetics.

### 2.10. FESEM and EDS Analysis

The surface morphologies and compositions of the PVA/PEI/GEN/TA0 and PVA/PEI/GEN/TA1 hydrogel samples were analyzed using FESEM and EDS techniques. Heterogeneous and regular structures were observed in the FESEM images (Appendix A), indicating successful sample preparation [74]. An elemental analysis of the sample surfaces (Appendix A) revealed the existence of C, O, and S elements in the PVA/PEI/GEN/TA0 sample, P and Ca elements originating from the SBF solution residue, and C, O, and S elements in the PVA/PEI/GEN/TA1 sample. The specific surface compositions of the samples can be found in Table 4.

### 2.11. Cell Viability Test

The cell viability assessment was performed by the WST-1 assay (Figure 9). The results revealed that both PVA/PEI/GEN/TA0 and PVA/PEI/GEN/TA1 caused a reduction in the viability of the L929 cells. However, PVA/PEI/GEN/TA1 exhibited a higher cell viability percentage compared to PVA/PEI/GEN/TA0. The L929 cell viability notably decreased to 56.6%, 51.3%, 50.1%, 48.5%, and 72.2% at various dilutions of PVA/PEI/GEN/TA0 (1:1, 1:2, 1:4, 1:10, and 1:100, respectively) (*p* < 0.01). In contrast, the treatment with PVA/PEI/GEN/TA1 showed lower cell toxicity (70.7%, 63.9%, 61.8%, 62.7%, and 111.0% at 1:1, 1:2, 1:4, 1:10, and 1:100, respectively). Especially, the 1:100 dilution of PVA/PEI/GEN/TA1 did not have any toxic effects on L929 cells due to a higher viability rate than the control group. Consequently, it can be inferred that tannic acid can mitigate the toxicity induced by gentamicin in the cells.

## 3. Conclusions

This study describes the successful synthesis of gentamicin-loaded and non-gentamicin-loaded TA-doped PVA-PEI hydrogel samples via the freeze–thaw method to investigate their potential as wound dressings for wound care and repair with high antibacterial properties. FTIR analysis indicates crosslinking between PVA and PEI polymers in the presence of TA. The hydrogel samples demonstrate favorable swelling and moisture retention capacities. Both drug-loaded and non-drug-loaded samples exhibit strong antibacterial properties, which is consistent with the literature reporting PEI polymer’s effectiveness. Additionally, the swelling and drug-release kinetics of the developed hydrogels were examined using various kinetic equations. While the swelling of hydrogels occurred in accordance with the pseudo-second-order equation and the Peleg model, GEN release from hydrogels was found to be diffusion-controlled and follows the Fickian diffusion mechanism, which fits the Higuchi and Korsmeyer–Peppas models.

Furthermore, gentamicin-doped hydrogel samples display a wider effective area, demonstrating their improved properties. The hydrogels offer 48 h of antibacterial protection, making them advantageous for use as wound dressings. Contact angle measurements confirm the samples’ desirable hydrophilic properties. Examination of the microstructure and morphology shows that the products have successfully attained regular and irregular porous structures. Elemental analysis of the hydrogel structures reveals the presence of C, O, P, S, and Ca elements. The WST-1 cytotoxicity test indicates that the TA additive reduces the cytotoxic effects of the samples on L929 cells. These findings highlight the potential for these hydrogels to provide controlled drug release, moisture, and oxygen permeability, with long-term and high antibacterial effects, positioning them as potential burn wound dressings.

## 4. Materials and Methods

### 4.1. Polymers and Additive Materials

In the presented investigation, a hydrogel was synthesized by incorporating polyvinyl alcohol (Merck, Darmstadt Germany) and polyethyleneimine (Sigma-Aldrich, St. Louis, MO, USA) polymers, with the crosslinking agent being tannic acid (Merck Brand). The hydrogel was enriched with gentamicin (40 mg/mL injection solution), known for its potent antibacterial activity to enhance its antibacterial characteristics further.

The preparation of the phosphate buffer solution (PBS) involved utilizing disodium hydrogen phosphate (Na_2_HPO_4_, Sigma-Aldrich) and monosodium phosphate (NaH_2_PO_4_) salts, with the pH being adjusted using hydrochloric acid (HCl) to attain the desired balance [75].

For the creation of simulated body fluid (SBF), ion-exchanged and distilled water (DW) were combined with various salts, including NaCl, NaHCO_3_, KCl, K_2_HPO_4_·3H_2_O, MgCl_2_·6H_2_O, CaCl_2_, and NaSO_4_. In addition, Tris-hydroxymethyl aminomethane (Tris) and 1 M HCl, as well as pH standard solutions (pH 4, 7, and 9), were employed in this process.

Furthermore, the WST-1 cytotoxicity testing utilized mouse fibroblasts, and the DMEM environment (Gibco; Thermo Fisher Scientific, Inc., Waltham, MA, USA) was filled with 10% fetal bovine serum (Gibco) and 1% penicillin-streptomycin (Gibco).

### 4.2. Devices and Equipment

The hydrogel solutions were prepared using a magnetic stirrer subjected to a high-temperature magnetic field. The pH of the PBS was adjusted via a Mettler Toledo pH meter. A standard graph for controlled drug-release analysis was established by mixing various concentrations of the GEN drug with an orbital shaker (Biosan, Rīga, Latvia). Characterization analyses were conducted using advanced equipment, including a Perkin Elmer Spectrum (Motic, Waltham, MA, USA) Two FT-IR Spectrometer (Metoree, Billerica, MA, USA), an optical microscope from Motic (Waltham, MA, USA), a thermal camera from FLIR (Teledyne FLIR, Wilsonville, OR, USA), a lyophilizer from Biobase (Jinan, China), an Attension contact angle measuring device (Nanoscience Instruments, Phoenix, AZ, USA), and a Shimadzu UV-2600 UV-Vis spectrophotometer (Shimadzu, Kyoto, Japan) for drug-release measurement. Furthermore, the hydrogels were incubated at 37 °C in a bacterial medium using a Nuve, FN 120 oven (Nuve, Tokyo, Japan). The mechanical properties were assessed using the Zwickroell tensile tester (Zwickroell, Guangzhou, China), and imaging was carried out using the Philips XL30 SFEG SEM (regen microscopy, Los Angeles, CA, USA) with WST-1 for cell quantification, Quanta FESEM, and the EVOS FL Cell Imaging System from Thermo Fisher Scientific (Waltham, MA, USA).

### 4.3. Preparation of Hydrogel

The drug-free hydrogels were prepared as follows. First, the polymer solutions were prepared separately. For this, 5 g of PVA were dissolved in 50 mL of DW by magnetic stirring at 80 °C for 2 h, and 2 g of PEI were dissolved in 10 mL of DW. Subsequently, the PEI solution was slowly added to the PVA solution at a 5:1 ratio, as outlined in Figure 10A, and mixed for 10 min. A portion of the resulting solution (10 mL) was transferred to a Petri dish to form the first hydrogel sample, while the remainder was distributed into separate 10 mL beakers. Various percentages of (1–6%) of TA were added to these beakers, and the mixtures were further blended for 15 min. The resulting solutions were then transferred to Petri dishes and subjected to the freeze–thaw method before being left to set in a refrigerator overnight to yield the PVA/PEI hydrogels.

Subsequently, a fresh PVA/PEI solution was prepared using the same steps and ratios, with the addition of GEN before distribution into Petri dishes (Figure 10C). A homogeneous 10 mL sample of the solution was then poured into a Petri dish, with the remaining solution divided into new 10 mL solutions containing increasing TA ratios and subsequently transferred to Petri dishes. Drug-containing PVA/PEI-based hydrogel samples were also prepared using the freeze–thaw method (Appendix A). These samples were readied for use in various characterization tests (Appendix A).

### 4.4. Preparation of PBS

The preparation of the phosphate-buffered saline (PBS) for conducting drug-release tests involved dissolving 1.7799 g of Na_2_HPO_4_ and 0.1913 g of NaH_2_PO_4_ salts in 50 mL of DW. Due to the rapid dissolution of the salts, 0.1 M HCl acid was added to adjust the pH of the alkaline solution to 7.4. Once the desired pH was achieved, the volume was adjusted to 100 mL using DW to obtain 0.1 M PBS.

### 4.5. Preparation of SBF

The SBF production followed the methodology outlined by Kokubo [76]. Special care was taken to ensure that the SBF was colorless and transparent. In a beaker, 700 mL of DW were heated to 36.5 ± 1.5 °C using a magnetic stirrer. Subsequently, specific amounts of NaCl, KCl, NaHCO_3_, K_2_HPO_4_·3H_2_O, CaCl_2_, MgCl_2_·6H_2_O, and Na_2_SO_4_, as indicated in Appendix A, were added in sequence to the heated water and dissolved. The total volume was adjusted to 900 mL. The pH of the solution was initially measured at 2. Tris reagent was then gradually added while monitoring the pH until it reached 7.3 at 36.5 ± 0.5 °C. A small excess of Tris was added, and the pH was allowed to stabilize. To adjust the pH to 7.4, if necessary, 1 M HCl was slowly introduced. Once the pH reached the desired level, the SBF was transferred to a 1 L volumetric flask and completed to 1 L with DW for further use.

### 4.6. FTIR Analysis

The study encompassed the evaluation of hydrogel specimens, specifically PVA/PEI/TA0, PVA/PEI/TA1, PVA/PEI/TA2, PVA/PEI/TA3, PVA/PEI/TA4, PVA/PEI/GEN/TA0, PVA/PEI/GEN/TA1, PVA/PEI/GEN/TA2, PVA/PEI/GEN/TA3, and PVA/PEI/GEN/TA4. FTIR analyses were performed by a Perkin Elmer (Spectrum Two) spectrometer using samples cut in a cylindrical shape with a thickness of 0.5 mm from hydrogels that were subjected to freeze–thaw cycles without being lyophilized. The assessment encompassed configuring the % transmittance mode within the wavenumber range of 400–4000 cm^−1^ to capture the FTIR spectra.

### 4.7. Swelling Test

The cylindrical-shaped PVA/PEI/GEN/TA0, PVA/PEI/GEN/TA1, PVA/PEI/GEN/TA2, PVA/PEI/GEN/TA3, and PVA/PEI/GEN/TA4 hydrogel samples with a diameter of 0.8 cm were cut, and their initial weights (*A*_0_) were measured using a precision balance. Subsequently, beakers were filled with 20 mL of distilled water, and the prepared samples were individually placed in the beakers. At predetermined time intervals, each sample was taken out of the water, excess water was removed, and their weights were recorded (*A*). This process was repeated at specified time intervals over a 72 h period. Additionally, the measurement results were calculated using the formula [77].
(14)$$\mathrm{Swelling}\;\mathrm{ratio}\;\%=\frac{A-A_{0}}{A_{0}}\times 100$$

### 4.8. Water Retention Test

Cylindrical samples with a diameter of 0.5 cm were prepared and initial measurements (*A*_0_) of the hydrogels were taken using a precision balance. Subsequently, the hydrogel samples were placed on a tray and allowed to air-dry at 37 °C. The masses of the samples (*A*) were periodically measured over a 432 h duration. The moisture retention of the samples was then calculated using the prescribed formula [78].
(15)$$\mathrm{Water}\;\mathrm{retention}\;\%=\frac{A}{A_{0}}\times 100$$

### 4.9. Contact Angle Measurements

The measurement of contact angles is utilized to assess the hydrophilic or hydrophobic characteristics of hydrogel samples. The commonly acknowledged criteria state that samples possessing a contact angle below 90° are hydrophilic, whereas those with a contact angle higher than 90° are hydrophobic [79]. Distilled water was meticulously loaded into a Hamilton syringe for this assessment. Subsequently, cylindrical PVA/PEI/TA0, PVA/PEI/TA1, PVA/PEI/TA2, PVA/PEI/TA3, PVA/PEI/TA4, PVA/PEI/GEN/TA0, PVA/PEI/GEN/TA1, PVA/PEI/GEN/TA2, PVA/PEI/GEN/TA3, and PVA/PEI/GEN/TA4 hydrogel samples, each with a diameter of 1.4 cm, were positioned in the camera focus of the measurement apparatus. A 4 μL droplet of distilled water from the syringe was then dispensed onto the samples, and their contact angles were computed using a computerized method.

### 4.10. Tensile Test

The test entails assessing the mechanical strength of the prepared hydrogel samples. Initially, the hydrogel samples were shaped into rectangles, and their cross-sectional areas (*S*) were computed. Subsequently, each sample was individually placed in the jaw of the device and subjected to tension by pulling from both ends. The force per unit applied during this process is known as tension. As per the literature, the results were calculated as MPa using the designated formula [80].
(16)$$\mathrm{Tensile}\;\mathrm{strength}=\frac{Force}{S}$$

### 4.11. SEM Analysis

The hydrogel samples, each with a 1.4 cm diameter, were placed in individual Petri dishes with the lids partially closed. They underwent freeze-drying at −20 °C for 24 h and an additional 12 h at −60 °C in a lyophilizer to ensure low-pressure lyophilization. After complete drying, the Petri dish lids were closed to prevent moisture ingress. Subsequently, the morphological analysis of the hydrogel samples was performed via a SEM specifically, the Philips XL30 SFEG model. Prior to testing, the samples were affixed to a metal base using conductive adhesive. Imaging was performed at a test voltage of 15 kV [81].

### 4.12. Antibacterial Test

The hydrogel samples were cut into 0.8 mm diameters and prepared for testing to evaluate their effects against *E. coli* and *S. aureus* bacteria, which are known to proliferate around wound tissue, causing inflammation. In the initial step of the testing process, the bacteria were seeded into the media and allowed to incubate briefly. Subsequently, the hydrogel samples were methodically introduced into the bacteria-seeded media within Petri dishes, which were then subjected to incubation at 37 °C for 24 and 48 h. This incubation aimed to assess whether the hydrogel samples exhibited any antibacterial properties.

### 4.13. Self-Healing Test

Test samples of the hydrogel were prepared by cutting them into cylindrical shapes with a diameter of 0.8 mm. To capture images, the samples were individually positioned on a slide under an inverted optical microscope (Motic BA310 Upright equipped with a digital camera, Richmond, BC, Canada), and images were taken at 40×/0.65/S (WD 0.5 mm) magnification. A photograph was taken of the initial state of each sample. Subsequently, a precise scratch was made on the hydrogel sample using a sharp, pointed razor blade, and the resulting image was promptly captured and recorded. After 5 min had elapsed since the initial scratch, a second set of images was taken, followed by a third set 15 min later. These images were obtained to assess the self-healing capability and to determine if the hydrogel sample scratch had dissipated [82].

### 4.14. In Vitro Drug-Release Studies

The initial step involved diluting GEN at varying concentrations and subsequently agitating it for 15 min on an orbital shaker (Biosan, Rīga, Latvia) at a speed of 200 rpm. A standard graph for GEN was then established through UV-Vis measurements, which were used for accurate quantification. Individual PVA/PEI/GEN/TA0, PVA/PEI/GEN/TA1, PVA/PEI/GEN/TA2, PVA/PEI/GEN/TA3, and PVA/PEI/GEN/TA4 hydrogel samples, each containing GEN, were placed in separate Eppendorf tubes with 2 mL of 0.1 M PBS (pH 7.4). The concentration of GEN in the 0.6 mL samples obtained from the tubes at specified time intervals was determined using UV-Vis measurements against the 0.1 M PBS standard solution, and the quantities of GEN were calculated based on the previously established standard graph. After each sampling, 0.6 mL of fresh 0.1 M PBS was introduced into the Eppendorf tubes [83,84].

### 4.15. FESEM-EDS Analysis

The hydrogel samples underwent surface morphology, smoothness, and surface composition analyses. The process involved freezing the samples at −20 °C for 12 h and lyophilization (Biobase) at −50 °C at 15 Pa for 2 days. Subsequently, the samples were immersed in a prepared SBF solution for 7 days and subjected to FESEM-EDS analyses at a voltage of 15 kV.

### 4.16. WST-1 Test

Mouse L929 fibroblasts were obtained from Dr. Secil Ak Aksoy and cultured in a DMEM medium (Gibco; Thermo Fisher Scientific, Inc., Waltham, MA, USA) filled with 10% fetal bovine serum (Gibco) and 1% penicillin and streptomycin (Gibco) in a 5% CO_2_ incubator. The cytotoxic effects of the hydrogels were analyzed using the dilution WST-1 test method [85]. First, the cells were seeded at 2 × 10^4^/well in a 96-well microplate and incubated for 24 h. The 100 mg gel samples were sterilized by UV and swollen in PBS. Then, the UV-sterilized hydrogels were added into 10 mL of culture medium and incubated at 37 °C for 24 h. Afterward, the cells were maintained with a serial dilution of the hydrogels with DMEM medium for 24 h. Afterward, 10 µL WST-1 dye (Biovision, San Francisco, CA, USA) was suffixed into each well and analyzed by an absorbance reader at 450 nm. Each experiment was repeated three times. Cell viability was accepted as the mean percentage of viable cells compared with the control group. The control group was regarded as 100% viable.

### 4.17. Statistical Evaluation

Calculating the means and standard deviations (SD) of the data was presented. The statistical comparison of the data was achieved by SPSS version 22 (IBM Corp., Armonk, NY, USA). A one-way variance analysis post-Tukey analysis was performed to compare the groups. *p*-values of <0.05 were regarded as significant. The percentage of L929 cell viability upon treatment with a dilution series of PVA/PEI/GEN/TA0 and PVA/PEI/GEN/TA1 for 24 h (*p* < 0.05 *, *p* < 0.01 **) was determined.

## Figures and Tables

**Figure 1 gels-10-00682-f001:**
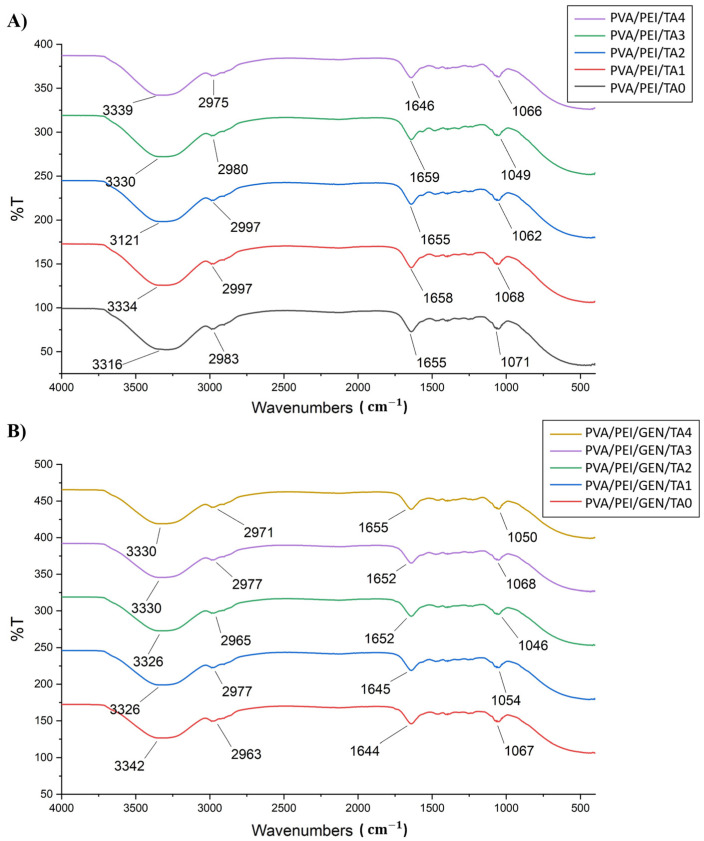
FTIR analysis of PVA/PEI-based hydrogel samples (**A**) without the addition of the drug and (**B**) with the addition of the drug.

**Figure 2 gels-10-00682-f002:**
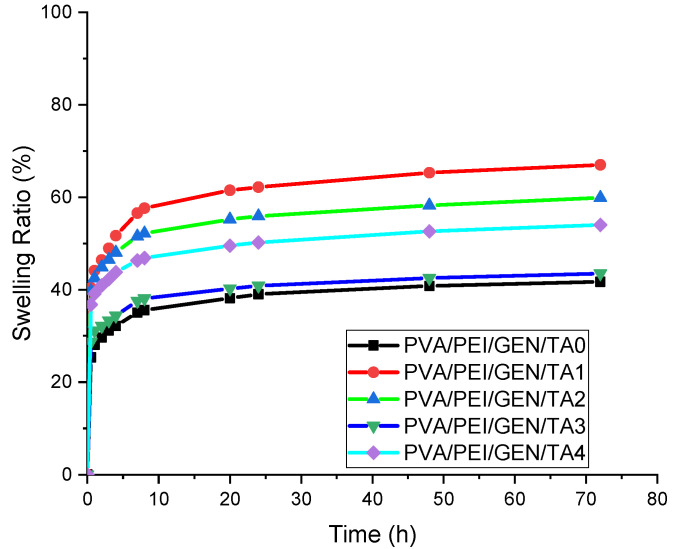
Swelling ratios of PVA/PEI-based hydrogel samples.

**Figure 3 gels-10-00682-f003:**
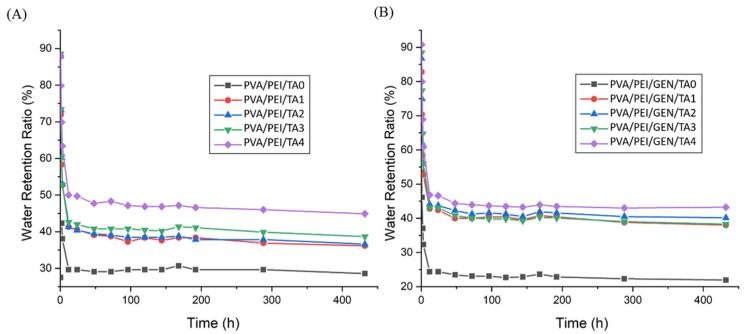
Water retention rate graph of PVA/PEI-based hydrogel samples. (**A**) Without the addition of the drug (**B**) With the addition of the drug.

**Figure 4 gels-10-00682-f004:**
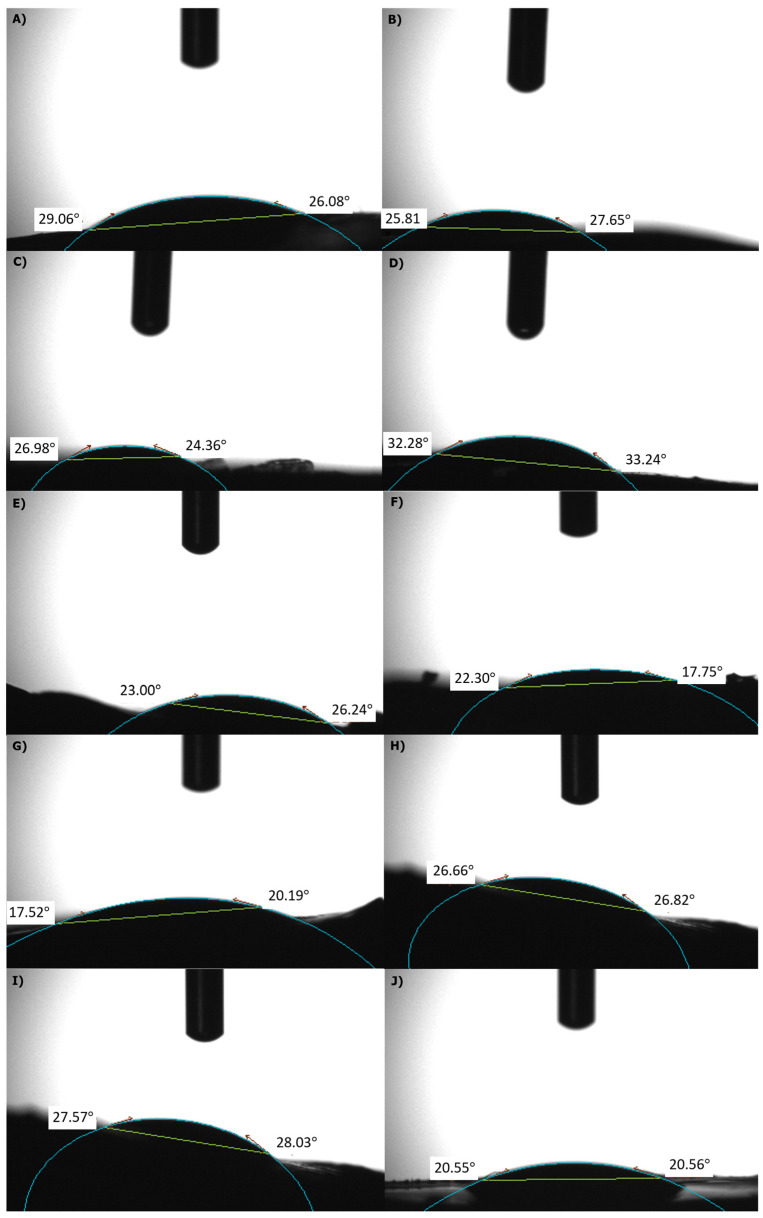
Contact angle measurements of PVA/PEI based hydrogels (**A**) PVA/PEI/GEN/TA0, (**B**) PVA/PEI/GEN/TA1, (**C**) PVA/PEI/GEN/TA2, (**D**) PVA/PEI/GEN/TA3, (**E**) PVA/PEI/GEN/TA4, (**F**) PVA/PEI/TA0, (**G**) PVA/PEI/TA1, (**H**) PVA/PEI/TA2, (**I**) PVA/PEI/TA3, (**J**) PVA/PEI/TA4.

**Figure 5 gels-10-00682-f005:**
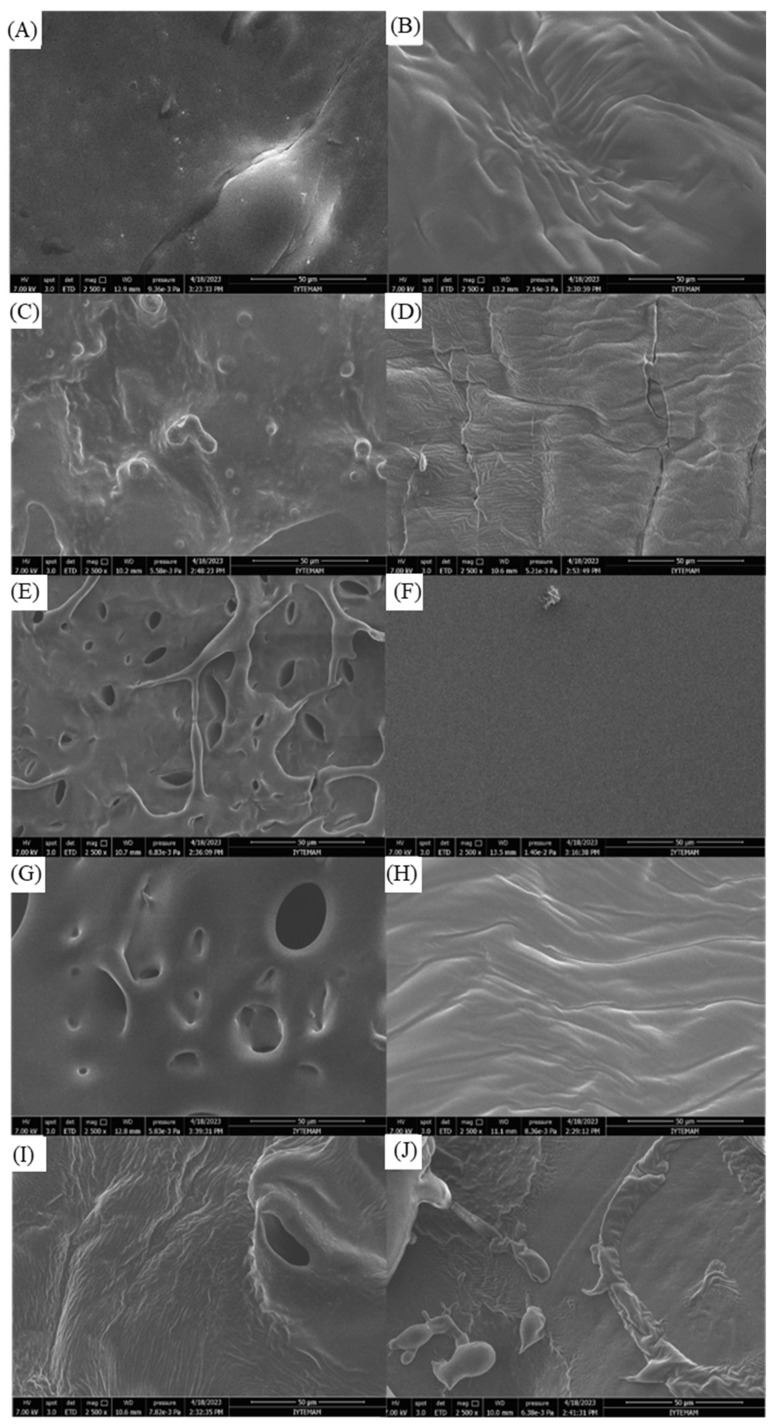
SEM micrographs of PVA/PEI-based hydrogels with and without drug addition. (**A**) PVA/PEI/GEN/TA0, (**B**) PVA/PEI/GEN/TA1, (**C**) PVA/PEI/GEN/TA2, (**D**) PVA/PEI/GEN/TA3, (**E**) PVA/PEI/GEN/TA4, (**F**) PVA/PEI/TA0, (**G**) PVA/PEI/TA1, (**H**) PVA/PEI/TA2, (**I**) PVA/PEI/TA3, (**J**) PVA/PEI/TA4.

**Figure 6 gels-10-00682-f006:**
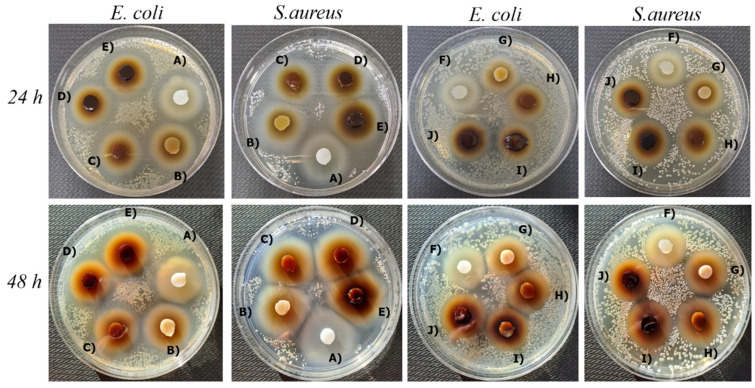
Antibacterial activity areas of PVA/PEI-based hydrogels with and without drug addition for 24 and 48 h. (A) PVA/PEI/GEN/TA0, (B) PVA/PEI/GEN/TA1, (C) PVA/PEI/GEN/TA2, (D) PVA/PEI/GEN/TA3, (E) PVA/PEI/GEN/TA4, (F) PVA/PEI/TA0, (G) PVA/PEI/TA1, (H) PVA/PEI/TA2, (I) PVA/PEI/TA3, (J) PVA/PEI/TA4.

**Figure 7 gels-10-00682-f007:**
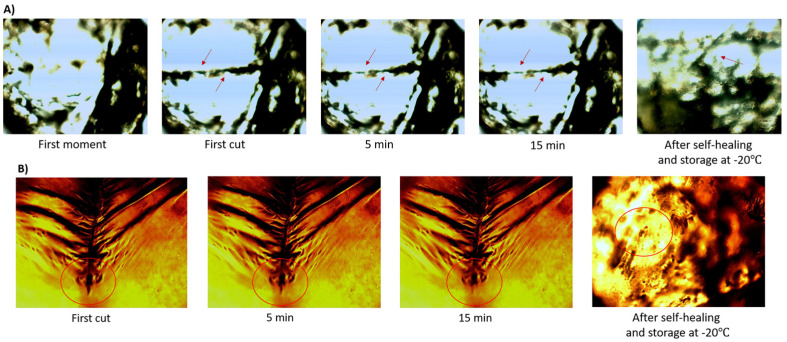
Images of self-healable hydrogel samples (**A**) PVA/PEI/TA0 and (**B**) PVA/PEI/TA1. Red arrows and circles indicate the self-healing regions of the hydrogels after scratches were made on the hydrogels.

**Figure 8 gels-10-00682-f008:**
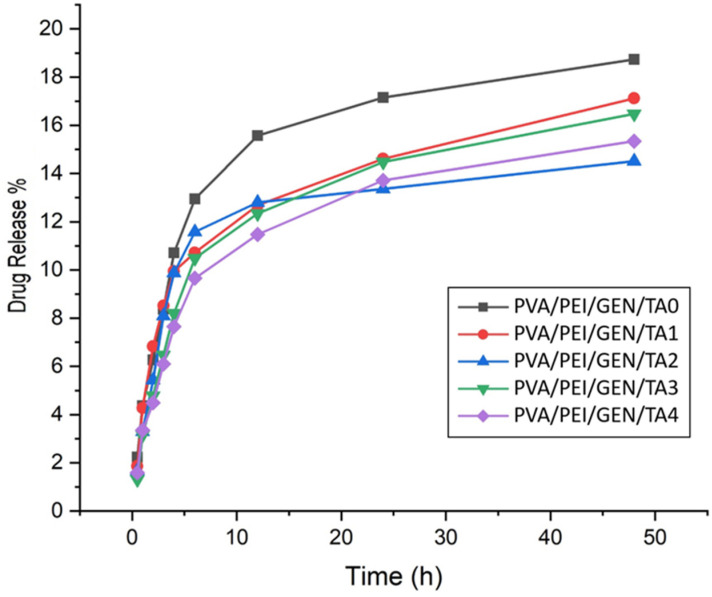
Drug release of PVA/PEI/GEN/TA hydrogel samples with gentamicin added.

**Figure 9 gels-10-00682-f009:**
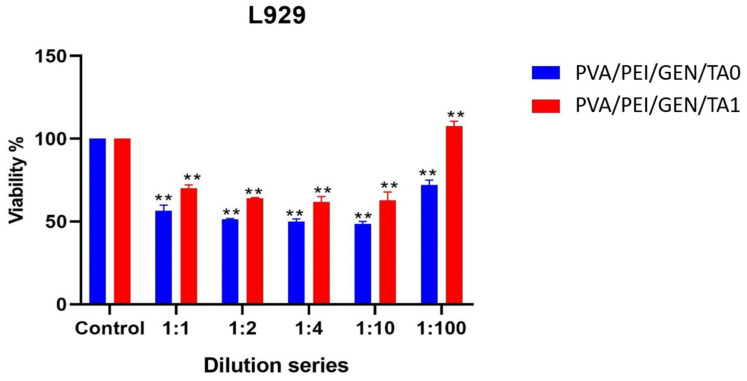
The percentage of L929 cell viability upon treatment with dilution series of PVA/PEI/GEN/TA0 and PVA/PEI/GEN/TA1 for 24 h (*p* < 0.01 **).

**Figure 10 gels-10-00682-f010:**
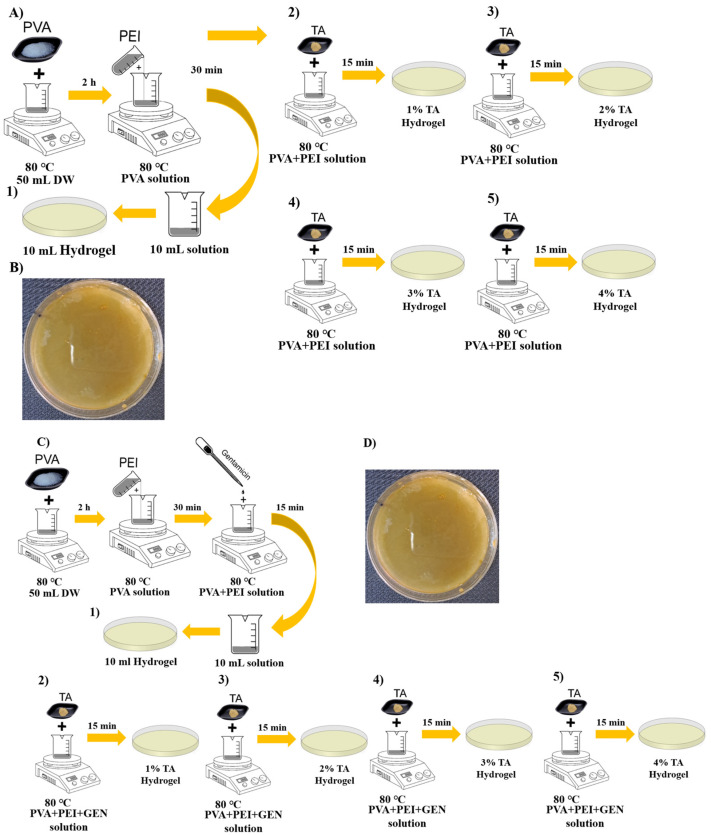
Illustrations outlining the preparation procedures and specimens of TA-added PVA/PEI hydrogel. (**A**) Specimens lacking drug addition. (**B**) PVA/PEI-based hydrogel specimen devoid of drug addition. (**C**) Specimens incorporating drug addition. (**D**) PVA/PEI-based hydrogel specimen including drug addition.

**Table 1 gels-10-00682-t001:** Swelling kinetic parameters for TA-doped PVA/PEI-based hydrogels.

Sample	Peppas Model			Peleg Model			Pseudo-First Order		Pseudo-Second Order	
	n	k_P_	r^2^	k_1_	k_2_	r^2^	k_1_	r^2^	k_2_	r^2^
PVA/PEI/GEN/TA0	0.1013	1.495	0.9841	0.1881	0.0495	0.9985	0.0650	0.8575	7.652	0.9994
PVA/PEI/GEN/TA1	0.1054	1.515	0.9761	0.0617	0.0249	0.9985	0.0616	0.8328	4.569	0.9994
PVA/PEI/GEN/TA2	0.0837	1.403	0.9869	0.0611	0.0293	0.9982	0.0566	0.7759	6.188	0.9994
PVA/PEI/GEN/TA3	0.0861	1.412	0.9798	0.1538	0.0495	0.9980	0.0612	0.8081	8.147	0.9995
PVA/PEI/GEN/TA4	0.0782	1.380	0.9936	0.0763	0.0342	0.9977	0.0569	0.7868	6.942	0.9993

**Table 2 gels-10-00682-t002:** Mechanical strength of TA-doped PVA/PEI-based hydrogels.

Samples	MPa	Samples	MPa
PVA/PEI/TA0	0.125	PVA/PEI/GEN/TA0	0.71
PVA/PEI/TA1	0.730	PVA/PEI/GEN/TA1	0.764
PVA/PEI/TA2	0.046	PVA/PEI/GEN/TA2	0.076
PVA/PEI/TA3	0.0373	PVA/PEI/GEN/TA3	-
PVA/PEI/TA4	-	PVA/PEI/GEN/TA4	-

**Table 3 gels-10-00682-t003:** Kinetic parameters for GEN releasing of TA-doped PVA/PEI based hydrogels.

Samples	Zero Order		First Order		Higuchi		Korsmeyer–Peppas			Hixson–Crowell	
	k_0_	r^2^	k_1_	r^2^	k_H_	r^2^	k_KP_	n	r^2^	k_HC_	r^2^
PVA/PEI/GEN/TA0	0.2975	0.6412	0.0013	0.7108	2.610	0.8349	0.2366	0.4551	0.8979	0.0769	0.4740
PVA/PEI/GEN/TA1	0.7834	0.9742	0.0010	0.5347	2.214	0.8610	0.2436	0.4357	0.8595	0.0782	0.6033
PVA/PEI/GEN/TA2	0.2128	0.5206	0.0013	0.7233	1.932	0.7261	0.2541	0.4562	0.8270	0.0736	0.5272
PVA/PEI/GEN/TA3	0.8104	0.9914	0.0012	0.7337	2.393	0.8813	0.1850	0.5256	0.8943	0.0839	0.6206
PVA/PEI/GEN/TA4	1.7272	0.9947	0.0012	07337	2.195	0.8914	0.2079	0.4805	0.9196	0.0855	0.7311

**Table 4 gels-10-00682-t004:** Elemental analysis of PVA/PEI/GEN/TA0 and PVA/PEI/GEN/TA1 hydrogel samples.

Hydrogel	Element	Weight %	Atamic %	Net Int.
PVA/PEI/GEN/TA0	C	57.29	65.51	66.94
	O	37.66	32.33	41.48
	S	5.05	2.16	20.07
PVA/PEI/GEN/TA1	C	53.26	61.72	102.2
	O	41.56	36.16	70.43
	P	1.29	0.58	7.60
	S	2.19	0.95	12.79
	Ca	1.70	0.59	5.66

## Data Availability

The original contributions presented in the study are included in the article, further inquiries can be directed to the corresponding authors.

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
