# Peer review of "Investigation of Tannic Acid Crosslinked PVA/PEI-Based Hydrogels as Potential Wound Dressings with Self-Healing and High Antibacterial Properties"

_gels, 2024, doi:10.3390/gels10110682_

Round 1
Reviewer 1 Report
Comments and Suggestions for Authors Karakus and co-workers report on the fabrication and properties of tannic acid cross-linked PVA/PEI-based hydrogels. Such macrogels are interesting for wound dressing and the authors show that the gels have excellent
antibacterial properties. This is even true for the problematic S. aureus. Hence, the present work shows that such gels should be further explored in the context of wound dressings.
This is an interesting topic and the results seem to be promising at least in part.
However, many of the plots are only presented without really making use of them. The authors could study swelling kinetics and also release kinetics. At the moment this is only done in very qualitative way. Moreover, I would completely delete the Li+ salt part. When I got this right the LiCl concentration which was used was 1Mol per liter. However, if this would be incorporated by the patient, this would be toxic. At 25 mg/l in the bloodstream the lethal dose is reached. Hence, I do not see any sense in adding lithium salts.
Besides these points the following things should be addressed in a revision:
-line 2-3: ...Hydrogels as Potential Wound Dressings .....
- The Abstract is by far too long. Please shorten this significantly.
-line 49: ...by bacteria.
-line 55-56: ...without dissolving. or: ....without dissolution.
-line 57: Delete the first sentence.
-line 88: What do you mean with "radiation"? Irradiation (e.g. UV cross-linking?)
-line 138: I would replace "betwixt" by "between"
-Fig. 1: Did you freeze dry the samples? Also in the methods section nothing is said about this. Is not, the water content does not allow any conclusion about the polymers OH groups.
-line 159: ...with drug addition.
-line 161: ....the cross-linked sample....
-Fig. 2: Plot only data up to 75h. The rest has no additional information since the plateau is already reached. Do you have any idea how the swelling kinetics can be modeled? Is this diffusion controlled?
-line 179: ...sample is promising for clinical studies...
-lines 187-190: I do not exactly understand what this means. Can you pleas try to rephrase this.
-Figure 5: Scale bars are to small and can't be read. Even zooming into the images does not help because the resolution of the images is too low for that. Hence, please indicate the lengthscales in the caption.
Why do you use SEM here. Optical differential phase contrast microscopy would probably do the same or a better job. Why don't you increase magnification to 1 micrometer scale or better? The presented images do not reveal anything. Also the working distance and the type of microscope can not be read.
-Fig. 7: How did you take these images? What are imaging parameters here?
-Fig. 8: What is this? This does not reveal anything without explanation?
-Fig 9: Can this be modeled by a release model? E.g. Peleg's Kinetic model? Or something else? Determination of the kinetic constant would maybe allow systematic comparison of the gels.
-Fig. 10: The sale bar is 100 micrometer. This is maybe ok for the interaction with cells. However, if a relation between drug release and structure shall be established 100 nm scale would be much more interesting.
-The section about cell viability is incomprehensible for me. Are the differences really significant. How did you obtain the error bars? What does the percentage 111 mean?
-Please re-write the first sentence of the conclusion.
Comments on the Quality of English Language
English is basically OK.
Author Response
Comments and Suggestions for Authors
Karakus and co-workers report on the fabrication and properties of tannic acid cross-linked PVA/PEI-based hydrogels. Such macrogels are interesting for wound dressing and the authors show that the gels have excellent antibacterial properties. This is even true for the problematic S. aureus. Hence, the present work shows that such gels should be further explored in the context of wound dressings.
This is an interesting topic and the results seem to be promising at least in part.
The authors thank the reviewer#1 for reviewing the manuscript and for all the interesting comments. All the changes are highlighted in blue font in the manuscript.
However, many of the plots are only presented without really making use of them. The authors could study swelling kinetics and also release kinetics. At the moment this is only done in very qualitative way. Moreover, I would completely delete the Li+ salt part. When I got this right the LiCl concentration which was used was 1Mol per liter. However, if this would be incorporated by the patient, this would be toxic. At 25 mg/l in the bloodstream the lethal dose is reached. Hence, I do not see any sense in adding lithium salts.
Response: Swelling and drug release kinetics were examined and added to the revised manuscript. Also, the parts related to LiCl were removed from the manuscript.
Besides these points the following things should be addressed in a revision:
-line 2-3: ...Hydrogels as Potential Wound Dressings .....
Response: The suggested correction has been made (lines 2-3).
- The Abstract is by far too long. Please shorten this significantly.
Response: Abstract has been shortened.
-line 49: ...by bacteria.
Response: The suggested correction has been made (line 41).
-line 55-56: ...without dissolving. or: ....without dissolution.
Response: The suggested correction has been made (line 47).
-line 57: Delete the first sentence.
Response: The first sentence has been deleted (line 49).
-line 88: What do you mean with "radiation"? Irradiation (e.g. UV cross-linking?)
Response: The "radiation" has been corrected as "UV irradiation" (line 80).
-line 138: I would replace "betwixt" by "between"
Response: The "betwixt" has been corrected as "between" (line 157).
-Fig. 1: Did you freeze dry the samples? Also in the methods section nothing is said about this. Is not, the water content does not allow any conclusion about the polymers OH groups.
Response: Hydrogels prepared with freeze-thaw cycles were used in FTIR analyses. This explanation was also added to the methods section (line 540). This part of manuscript is explained in more detail by taking into account the also recommendations of Refeere#2.
-line 159: ...with drug addition.
Response: The sentence ”….the best swelling capacity is PVA/PEI/GEN/TA1 with drug addiction.” has been changed as ”….the best swelling capacity is drug-added PVA/PEI/GEN/TA1.” (line 178).
-line 161: ....the cross-linked sample....
Response: The statement ”…. the cross-linker sample….” has been changed as ”…. the cross-linked sample….” (line 180).
-Fig. 2: Plot only data up to 75h. The rest has no additional information since the plateau is already reached. Do you have any idea how the swelling kinetics can be modeled? Is this diffusion controlled?
Response: Figure 2 has been redrawn for data up to 72h. Also, the kinetic modeling has been added.
-line 179: ...sample is promising for clinical studies...
Response: The statement ”…. sample shows promise for clinical studies….” has been changed as ”…. sample is promising for clinical studies….” (line 259).
-lines 187-190: I do not exactly understand what this means. Can you pleas try to rephrase this.
Response: Necessary arrangements have been made for clearly understand (lines 267-274).
-Figure 5: Scale bars are to small and can't be read. Even zooming into the images does not help because the resolution of the images is too low for that. Hence, please indicate the lengthscales in the caption. Why do you use SEM here. Optical differential phase contrast microscopy would probably do the same or a better job. Why don't you increase magnification to 1 micrometer scale or better? The presented images do not reveal anything. Also the working distance and the type of microscope can not be read.
Response: Figure 5 has been rearranged so that scale bars can be seen.
-Fig. 7: How did you take these images? What are imaging parameters here?
Response: Explanations about imaging were added to the method section (lines 603-604).
-Fig. 8: What is this? This does not reveal anything without explanation?
Response: Since the parts related to LiCl have been removed from the manuscript, the antifreezing properties section and Figure 8 have also been omitted.
-Fig 9: Can this be modeled by a release model? E.g. Peleg's Kinetic model? Or something else?
Determination of the kinetic constant would maybe allow systematic comparison of the gels.
Response: Kinetic modeling was done for drug release and added to the manuscript.
-Fig. 10: The sale bar is 100 micrometer. This is maybe ok for the interaction with cells. However, if a relation between drug release and structure shall be established 100 nm scale would be much more interesting.
Response: No meaningful images could be obtained from hydrogel samples at 100 nm scale. Also, according to the opinion of referee#2, Figure 10 was removed from the manuscript and added to the Supplementary file.
-The section about cell viability is incomprehensible for me. Are the differences really significant. How did you obtain the error bars? What does the percentage 111 mean?
Response: The cell viability experiments for each dilution was performed at least three times. Therefore, the mean of three absorbance was summarized and difference among them summarized as standard deviation. The absorbance of the control group was accepted as 100% percentage and the other absorbance values were analyzed according to them. Therefore, the results were summarized percentage. Additionally, 111% means that the viability of the cells after treatment of 1:100 dilution, we obtained higher absorbance than the control. Therefore, the viability was higher than the control due to cell division. We added these parameters in the manuscript.
-Please re-write the first sentence of the conclusion.
Response: The first sentence of the conclusion has been rewritten.

Reviewer 2 Report
Comments and Suggestions for Authors
In this study, the researchers have developed a new hydrogel using a combination of polyvinyl alcohol (PVA) and polyethyleneimine (PEI) polymers for the treatment of burn wounds. Tannic acid (TA) was used for chemical crosslinking between the polymers. Additionally, the inclusion of lithium chloride (LiCl) provides the hydrogel with self-healing and non-freezing properties. Gentamicin (GEN) was incorporated into the hydrogel samples to create antibacterial wound dressings. Based on the quality of the journal, I do not believe it will be accepted.
The introduction should include a review of PVA/TA hydrogel-based dressings and emphasize the novelty of the submission. The data on the spectra in Figure 1 is difficult to read, and some data appears to be incorrect. Figure 2 needs modification, and the trend of swelling ratio with varying TA is unclear. More detailed explanations are needed at line 187. I also did not see any evidence that LiCl improved the moisture retention capacity. Figure 4 needs modification, and it's unclear how the extension ratio of the samples relates to Table 1. Figure 5 needs improvement, and the necessity for Figure 10 is unclear. After self-healing and storage at -20°C, the mechanical properties in Figure 7 should be studied. Furthermore, additional experiments are required to demonstrate the hydrogel's ability to promote wound healing.
Comments on the Quality of English Languagegood
Author Response
Responses to Comments and Suggestions for Authors
In this study, the researchers have developed a new hydrogel using a combination of polyvinyl alcohol (PVA) and polyethyleneimine (PEI) polymers for the treatment of burn wounds. Tannic acid (TA) was used for chemical crosslinking between the polymers. Additionally, the inclusion of lithium chloride (LiCl) provides the hydrogel with self-healing and non-freezing properties. Gentamicin (GEN) was incorporated into the hydrogel samples to create antibacterial wound dressings. Based on the quality of the journal, I do not believe it will be accepted.
The authors thank the reviewer#2 for reviewing the manuscript and for all the interesting comments. All the changes are highlighted in blue font in the manuscript.
- The introduction should include a review of PVA/TA hydrogel-based dressings and emphasize the novelty of the submission.
Response: A paragraph has been added in the introduction that includes a review of PVA/TA hydrogel-based dressings and highlights the novelty of this work (lines 101-118).
- The data on the spectra in Figure 1 is difficult to read, and some data appears to be incorrect.
Response: Comments regarding the data in Figure 1 were revised taking into account also referee #1's suggestions.
- Figure 2 needs modification, and the trend of swelling ratio with varying TA is unclear.
Response: Figure 2 was revised considering also referee #1's suggestion and an explanation was added regarding the relationship between the amount of TA and the swelling ratio (lines 181-191). Also, explanations were added regarding swelling kinetics.
- More detailed explanations are needed at line 187.
Response: More explanations have been added (lines 267-272).
- I also did not see any evidence that LiCl improved the moisture retention capacity.
Response: All explanations regarding LiCl were removed from the manuscript in accordance with the recommendation of Referee#1.
- Figure 4 needs modification, and it's unclear how the extension ratio of the samples relates to Table 1.
Response: Figure 4 is not well visible, so it has been corrected to be visible. Figure 4 shows the contact angle measurement results, and since it is not related to the mechanical tests, it has no relation to Table 1.
- Figure 5 needs improvement, and the necessity for Figure 10 is unclear.
Response: Figure 5 has been rearranged and Figure 10 has been transferred to the supplementary file.
- After self-healing and storage at -20°C, the mechanical properties in Figure 7 should be studied.
Response: After self-healing and storage at -20°C, new images were taken and added to Figure 7 and the relevant explanation was added to the manuscript (lines 344-345).
- Furthermore, additional experiments are required to demonstrate the hydrogel's ability to promote wound healing.
Response: Since our laboratory facilities do not have the means to test wound healing, additional experiments could not be conducted. We hope that this situation will be understood.

Reviewer 3 Report
Comments and Suggestions for Authors
This paper is a scientific experimental study conducted by the authors to develop an novel advanced antibacterial hydrogel wound dressing containing gentamicin, by the chemical crosslinking of polyvinyl alcohol (PVA) and polyethyleneimine (PEI) polymers in the presence of tannic acid (TA) and adding lithium chloride (LiCl) to the hydrogel solutions for gaining self-healing and non-freezing properties. Five gentamicin-loaded PVA/PEI based hydrogel samples were prepared, using 2% antibiotic, 1% LiCl, different TA concentrations (1, 2, 4 and 6%) and a PVA/PEI mixture in ratio of 5:1.
The main issue addressed by the research was reducing risks of infection, promoting wound healing, and supporting tissue repair in wound treatments by hydrogel wound dressings. The topic is original and relevant to the field because of the following reasons: the biocompatible combination of polymers used to obtain the hydrogels; the presence of tannic acid not only as crosslinking agent, but also for its synergic antibacterial effects with gentamicin; the use of LiCl to obtain self-healing and non-freezing properties.
All five hydrogel samples, including their corresponding controls, were tested for various characteristics as follow: Fourier transform infrared (FTIR) analysis to examine the desired functional groups of PVA/PEI hydrogels; tests for swelling abilities and moisture retention capacities; contact angle test to determine hydrophilic properties; tensile tests to assess mechanical strength; SEM and FESEM-EDS analyses to understand surface morphologies and elemental composition; antibacterial tests against E. coli and S. aureus bacteria to determine the materials' antimicrobial properties; cell viability analysis performed through WST-1 tests; scratch tests to evaluate the self-healing abilities; controlled drug release test (for drug-loaded hydrogel samples) and freezing resistance tests. All characterization tests were conducted using actual and appropriate methods.
The paper is original, is easy to read and understand, and has relevance for formulators, academics and healthcare practitioners using hydrogel wound dressings.
The title and abstract are appropriate for the content of the text, and the abstract is presented in a structured format. Furthermore, the article is well constructed and written, and treats an actual problem, which is clearly defined with appropriate reference to the literature.
However, the authors should clarify some aspects:
- lines 294-306, “2.10. Drug Release Profiles” subsection: The authors should present in more details the obtained results.
- lines 393-394, “4.3. Preparation of Hydrogel” subsection: Please specify the concentration of the used PVA and PEI solutions. Also, please specify the temperature of heating.
- line 399, “4.3. Preparation of Hydrogel” subsection: “Various percentages of PVA polymer weight were added…”. Probably it is a typing mistake, it should be TA instead of PVA.
- lines 452-453, “4.7. Swelling Test” subsection: “This process was 452 repeated at specified time intervals over a 432-hour period.”. Please explain why it was necessary a 432 hours period for this testing. Also, specify hydrogel samples submitted to this test.
- lines 522-533, “4.15. In vitro drug release studies” subsection: The described test protocol is not the commonly used one (based on vertical diffusion cells), so please indicate at least one reference for the chosen protocol.
Also, the macroscopic examination of the developed hydrogels (aspect, homogeneity, odor, consistency etc.) and their pH values are missing from the text of the manuscript. It is recommended to present these properties in the manuscript, since the tested hydrogels are intended for wound treatment.
Author Response
Responses to Comments and Suggestions for Authors
This paper is a scientific experimental study conducted by the authors to develop an novel advanced antibacterial hydrogel wound dressing containing gentamicin, by the chemical crosslinking of polyvinyl alcohol (PVA) and polyethyleneimine (PEI) polymers in the presence of tannic acid (TA) and adding lithium chloride (LiCl) to the hydrogel solutions for gaining self-healing and non-freezing properties. Five gentamicin-loaded PVA/PEI based hydrogel samples were prepared, using 2% antibiotic, 1% LiCl, different TA concentrations (1, 2, 4 and 6%) and a PVA/PEI mixture in ratio of 5:1.
The main issue addressed by the research was reducing risks of infection, promoting wound healing, and supporting tissue repair in wound treatments by hydrogel wound dressings. The topic is original and relevant to the field because of the following reasons: the biocompatible combination of polymers used to obtain the hydrogels; the presence of tannic acid not only ascrosslinking agent, but also for its synergic antibacterial effects with gentamicin; the use of LiCl to obtain self-healing and non-freezing properties.
All five hydrogel samples, including their corresponding controls, were tested for various characteristics as follow: Fourier transform infrared (FTIR) analysis to examine the desired functional groups of PVA/PEI hydrogels; tests for swelling abilities and moisture retention capacities; contact angle test to determine hydrophilic properties; tensile tests to assess mechanical strength; SEM and FESEM-EDS analyses to understand surface morphologies and elemental composition; antibacterial tests against E. coli and S. aureus bacteria to determine the materials' antimicrobial properties; cell viability analysis performed through WST-1 tests; scratch tests to evaluate the self-healing abilities; controlled drug release test (for drug-loaded hydrogel samples) and freezing resistance tests. All characterization tests were conducted using actual and appropriate methods.
The paper is original, is easy to read and understand, and has relevance for formulators, academics and healthcare practitioners using hydrogel wound dressings.
The title and abstract are appropriate for the content of the text, and the abstract is presented in a structured format. Furthermore, the article is well constructed and written, and treats an actual problem, which is clearly defined with appropriate reference to the literature.
However, the authors should clarify some aspects:
The authors thank the reviewer#3 for reviewing the manuscript and for all the interesting comments. All the changes are highlighted in blue font in the manuscript.
- lines 294-306, “2.10. Drug Release Profiles” subsection: The authors should present in more details the obtained results.
Response: The results obtained are presented in more detail (lines 355-362). Additionally, drug release kinetics were added considering also referee #1's suggestion.
- lines 393-394, “4.3. Preparation of Hydrogel” subsection: Please specify the concentration of the used PVA and PEI solutions. Also, please specify the temperature of heating.
Response: The concentrations of the used PVA and PEI solutions and the heating temperature were specified (lines 492-494).
- line 399, “4.3. Preparation of Hydrogel” subsection: “Various percentages of PVA polymer weight were added…”. Probably it is a typing mistake, it should be TA instead of PVA.
Response: PVA misspelling corrected to TA (line 498).
- lines 452-453, “4.7. Swelling Test” subsection: “This process was 452 repeated at specified time intervals over a 432-hour period.”. Please explain why it was necessary a 432 hours period for this testing. Also, specify hydrogel samples submitted to this test.
Response: The time for swelling test was changed to 72 hours as per referee#1's suggestion. Samples used for swelling testing are indicated (line 550).
- lines 522-533, “4.15. In vitro drug release studies” subsection: The described test protocol is not the commonly used one (based on vertical diffusion cells), so please indicate at least one reference for the chosen protocol.
Also, the macroscopic examination of the developed hydrogels (aspect, homogeneity, odor, consistency etc.) and their pH values are missing from the text of the manuscript. It is recommended to present these properties in the manuscript, since the tested hydrogels are intended for wound treatment.
Response: Two references regarding the test protocol have been added to the manuscript and are provided below. Also, macroscopic images of the hydrogels are given in Figure S6 (Supplementary file).
- Balcı, S., Camcı, Y., Türk, S., Altınsoy, İ., Çelebi Efe, G., Ipek, M., Özacar, M., Bindal, C., Ultrasound sensitive smart polyvinyl alcohol/melamine/tannic acid hydrogel. Arab. J. Sci. Eng. 2024, 49, 9221–9233.
- Türk, S., Altınsoy, İ., Çelebi Efe, G., Ipek, M., Özacar, M., Bindal, C., A novel multifunctional NCQDs-based injectable self-crosslinking and in situ forming hydrogel as an innovative stimuli responsive smart drug delivery system for cancer therapy. Mater. Sci. Eng. C 2021, 121, 111829.

Round 2
Reviewer 1 Report
Comments and Suggestions for Authors
The authors have improved their manuscript according to my first report. Now they make use of the data and I think the work will be of interest for the community.
Comments on the Quality of English LanguageEnglish is OK for me.
Reviewer 2 Report
Comments and Suggestions for Authors
It can be accepted.